# Propranolol Suppresses the T-Helper Cell Depletion-Related Immune Dysfunction in Cirrhotic Mice

**DOI:** 10.3390/cells9030604

**Published:** 2020-03-03

**Authors:** Hung-Cheng Tsai, Chien-Fu Hsu, Chia-Chang Huang, Shiang-Fen Huang, Tzu-Hao Li, Ying-Ying Yang, Ming-Wei Lin, Tzung-Yan Lee, Chih-Wei Liu, Yi-Hsiang Huang, Ming-Chih Hou, Han-Chieh Lin

**Affiliations:** 1Division of Allergy and Immunology, Department of Medicine, Taipei Veterans General Hospital, Taipei 11217, Taiwan; hctsai7@vghtpe.gov.tw (H.-C.T.); thli3@ym.edu.tw (T.-H.L.); cwliu2@vghtpe.gov.tw (C.-W.L.); 2Faculty of Medicine, School of Medicine, National Yang-Ming University School of Medicine, Taipei 11217, Taiwan; cfhsu8@vghtpe.gov.tw (C.-F.H.); cchuang7@vghtpe.gov.tw (C.-C.H.); sfhuang6@vghtpe.gov.tw (S.-F.H.); yhhuang@vghtpe.gov.tw (Y.-H.H.); mchou@vghtpe.gov.tw (M.-C.H.); hclin@vghtpe.gov.tw (H.-C.L.); 3Division of General Medicine, Department of Medicine, Taipei Veterans General Hospital, Taipei 11217, Taiwan; 4Division of Clinical Skills Center, Department of medical education, Taipei Veterans General Hospital, Taipei 11217, Taiwan; 5Institute of Clinical Medicine, National Yang-Ming University School of Medicine, Taipei 11217, Taiwan; 6Division of Infection, Department of Medicine, Taipei Veterans General Hospital, Taipei 11217, Taiwan; 7Division of Allergy, Immunology, and Rheumatology, Department of Internal Medicine, Shin Kong Wu Ho-Su Memorial Foundation, Taipei 11217, Taiwan; 8Division of Gastroenterology and Hepatology, Department of Medicine, Taipei Veterans General Hospital, Taipei 11217, Taiwan; 9Institute of Public Health, School of Medicine, National Yang-Ming University School of Medicine, Taipei 11217, Taiwan; mwlin@ym.edu.tw; 10Molecular Pharmacology Laboratory of Chinese Medicine, Chang Gung Memorial Foundation, Linkou Branch 333, Taiwan; joyamen@mail.cgu.edu.tw

**Keywords:** cirrhosis-associated immune dysfunction, splenic β adrenergic receptor, Th-cell depletion

## Abstract

Bacterial translocation (BT) and splenomegaly contribute to cirrhosis-associated immune dysfunction (CAID) including T cell depletion, infection, and chronic inflammation. β-blockers have been reported to decrease BT and improve splenomegaly. This study explores the modulation of β1 and β2 adrenergic receptors (ADRB1/ADRB2) by propranolol treatment on the peripheral and splenic immune dysfunction of cirrhotic mice. In vivo experiments were performed in bile duct ligation (BDL)- and thioacetamide (TAA)-cirrhotic mice receiving two weeks of propranolol treatment. Acute effects of propranolol were evaluated in T-helper (Th) cells isolated from spleen of cirrhotic mice. Over-expression of β1 and β2 adrenergic receptors (ADRB1/ADRB2) in spleen and T lymphocytes was associated with high peripheral/splenic lipopolysaccharide binding protein levels. Moreover, a decrease in Th cells percentage, increase in Treg subset, and cytokines were accompanied by increased apoptosis, proliferation, and reduced white pulp hyperplasia in cirrhotic mice, which were counteracted by propranolol treatment. The Th-cell depletion, systemic inflammation, BT, and infection were improved by chronic propranolol treatment. Acute propranolol treatment inhibited apoptosis, Treg-conditioned differentiation, and promoted Th2-conditioned differentiation through ADRB-cyclic adenosine monophosphate (cAMP) signals in cirrhotic mice. In conclusion, suppression of ADRB1 and ADRB2 expressions in spleen and splenic T lymphocytes by acute and chronic propranolol treatment ameliorate systemic and splenic immune dysfunction in cirrhosis.

## 1. Introduction

In cirrhosis, T cell depletion plays a key role in the pathogenesis of cirrhosis-associated immune dysfunction [1,2]. In addition, bacterial translocation (BT)-related expansion of splenic lymphoid tissue and inflammation, as well as splenic sequestration results in T lymphopenia [3,4,5,6]. BT is commonly found in cirrhotic patients with portal hypertension (PH), splenomegaly, and immune dysfunction.

Munoz et al. [7] have reported that in cirrhosis, over-activated T lymphocytes, and inflamed mesenteric lymph nodes (MLNs) facilitate BT and that such patients develop systemic inflammatory syndrome. MLNs and the spleen are closely connected via the portal vein. Similar to MLNs, BT to the spleen activates lymphocytes and cytokines release in the spleen. In physiological conditions, the spleen acts as a filter, in parallel to liver to eliminate bacterial antigens from both systemic and splanchnic circulation. In cirrhosis, splenomegaly has been considered as an activator of T cells and tissue inflammation due to a reduction in the hepatic capacity to handle BT-related pathogens [8]. Thus, both BT and splenomegaly contribute to cirrhosis-associated immune dysfunction [9,10].

Cirrhotic patients, especially in the de-compensated state show an increased activity of the sympathetic nervous system (SNS) [11,12]. In the spleen, the sympathetic nerve mainly innervates the T cells zone to regulate immune cell function. Both splanchnic and splenic nerves are sensitive to lipopolysaccharide-driven SNS activation. In cirrhotic rats, splanchnic SNS hyperactivity was found to cause translocation of *Escherichia coli* to MLNs and extra-intestinal sites [13].

A report indicated that propranolol increased intestine motility and improved portal hypertension; thus, decreasing the rate of BT in cirrhosis with ascites rats [14]. Additionally, in cirrhotic patients with acute-on-chronic liver failure, the use of β-blockers has been reported to decrease the incidence and severity of systemic inflammation, independent of hemodynamic response [15,16].

A β-adrenergic agonist, isoproterenol, inhibits the proliferation and differentiation of T cells [17]. In particular, T cell differentiation and activation are regulated by the β2-adrenergic receptors (ADRB2) expression on their surface [18,19]. Stimulation of the ADRB2 on naïve T cell populations favors Th1 differentiation, and this effect is blocked by β-antagonists. Overall, activation of ADRB2s inhibits cellular immunity. Therefore, in cirrhotic mice, activated hepatic β1-adrenergic receptor (ADRB1) facilitates BT, which can be counteracted by administering a pre-treatment with β1 blockers [11]. Propranolol treatment can suppress over-activated splenocytes, reduce inflammatory cytokines release, and improve splenomegaly in mice with social disruption stress [20].

Accordingly, this study aims to evaluate the roles of splenic ADRBs on the cirrhosis-associated T lymphopenia and immune dysfunction. In addition, the effects and mechanisms of chronic beta-blocker treatment on the above-mentioned abnormalities were evaluated.

## 2. Materials and Methods

### 2.1. Animals

C57BL/6 male mice were purchased from Charles River Japan, Inc. (Yokohama, Japan) and received humane care in accordance with the Guide for the Care and Use of Laboratory Animals (published by the National Institute of Health) in Yang-Ming animal facility. Two or three mice were kept in one cage. In addition to free access to food and water, the mice were provided a small block of wood for chewing and nesting material for resting. The health status of the mice was carefully examined throughout the experiments by veterinary assistants assigned to the Yang-Ming animal facility. The experiments were approved by the animal ethical committee of Yang-Ming medical university with approval of No. 1061008r.

Mice (8-week old) were treated with thioacetamide (TAA, 200 mg/kg) three times per week via intra-peritoneal injection for 16 weeks, to create cirrhotic mice with portal hypertension. Additionally, common bile duct ligation (BDL) was performed in mice to induce biliary cirrhosis, 5 weeks after BDL. Sham-operated (sham) mice had their bile ducts exposed but not ligated. Then, TAA- and BDL-cirrhotic mice were randomly assigned to receive oral gavage of either (S)-(−)-propranolol hydrochloride from Sigma (St. Louis, MO, USA; 30 mg/kg/day) or vehicle for 3 weeks. Thus, we created five experimental groups: sham-V, TAA-V, BDL-V, TAA-pro, and BDL-pro mice [*n* = 7, except in sham-V mice (*n* = 4)]. The administration of TAA was continued during the period of propranolol or vehicle treatment in TAA-V and TAA-pro groups. Approximately 10% of mice died during the process of induction of cirrhosis, by either BDL or TAA administration. The dose of propranolol was based on its capacity to block ADRBs and decrease portal pressure, in accordance to previous studies [20,21]. It had been reported that, in BDL-cirrhotic rats, 2 week of propranolol (30 mg kg^−1^ day^−1^) treatment can effectively decrease portal pressure by reducing portal blood flow, decreasing superior mesenteric artery (SMA) blood flow and neo-angiogenesis, suppressing hepatic fibrosis and neovascularization, and reducing portosystemic shunting [20]. In portal vein ligated rats, chronic propranolol (30 mg kg^−1^ day^−1^) treatment significantly alleviated the hyperdynamic state, including portal pressure, cardiac index, and total peripheral resistance and improved contractility of SMA [21]. Accordingly, this dose of duration of propranolol was used to test it effect on T cell dysfunction in cirrhotic mice. At the time of sacrifice, we evaluated the animals to look for the presence of ascites and to weight them using sterilized gauzes.

### 2.2. Ethical Approval

Approval for this study was obtained from the Institutional Ethics Review Committee of the University of Yang-Ming. All procedures were performed according to the guidelines of the Institutional Animal Care and Use Committee at the University of Yang-Ming, and the National Research Council’s Guide for the Care and Use of Laboratory Animals (1985). All procedures were performed according to the guidelines of the Institutional Animal Care and Use Committee at the University of Yang-Ming, and the National Research Council’s Guide for the Care and Use of Laboratory Animals (1985), DHEW Publication no. (NIH) 85-23: Office of Science and Health Reports, DRR/NIH, Bethesda, MD, USA.

### 2.3. Evaluation of Systemic Inflammation

Blood samples were taken by puncture of the aortic bifurcation and heart after portal pressure measurements and were assayed by automated procedures and commercially available ELISA kits. They were evaluated for plasma norepinephrine levels (Lifespan BioSciences, Seattle, WA, USA), levels of peripheral Th1 (IFN-γ and TNF-α), Th2 (IL-4 and IL-10), and Treg (TGFβ and IL-35) cytokines [mouse Quantikine ELISA kits (R&D Systems, Minneapolis, MN, USA)].

### 2.4. Detection of Systemic Infection

Using aerobic/anaerobic culture plates, sterile blood samples (1 mL), liver, lung, intestine, ascites, and pleural fluid (if available) were collected for evaluation of bacteremia and systemic infections, which is defined as any positive culture from the abovementioned biological samples.

### 2.5. Tissue Collections

Next, MLN, spleen, intestine, and liver were removed and collected immediately in liquid nitrogen and stored at −80 °C to halt the degradation of various pathogenic substances. Subsequently, ADRB1 and ADRB2 antibodies from Abcam (Cambridge, UK) were used to measure protein levels. To evaluate the impacts of β-blockers on systemic inflammatory syndrome in cirrhotic mice, the protein expressions of ADRB1 and ADRB2 in various immune dys-regulation-related tissues compared with sham mice. Our initial data revealed that simultaneous increase in ADRB1 and ADRB2 protein levels were noted only in the spleen rather than in the liver, intestine, and MLN. Thus, the following experiments focused on the spleen. Next, the percentage of CD3^+^ cells (representing T cells) among all splenocytes or ADRB1^+^ CD3^+^/ADRB2^+^ CD3^+^ cells among CD3^+^ splenocytes from different groups of mice were measured by immunofluorescence (IF) staining.

From each spleen, two sections (100 µm apart) were studied and from each section data four individual areas of white pulp (WP) were collected. The total area of lymphoid tissue was estimated by measuring WP area (%) in the spleen of each animal and averaged. Later, apoptosis in the spleen was measured by Terminal deoxynucleotidyl transferase dUTP nick end labeling (TUNEL) staining with an ApopTag Peroxidase In Situ Apoptosis Detection Kit obtained from Chemicon (Buena Park, CA, USA). The average percentage of TUNEL (fluorescence)-positive area was determined in five microscope fields (magnification, 40×) for each tissue sample to compare between groups. Digital images were analyzed using Image J Software. The levels of splenic lipopolysaccharide binding protein (LBP) were measured. Measurements were read in triplicates using a Tecan Sunrise automated microplate reader (Männedorf, Switzerland).

### 2.6. Preparation of Splenic T Lymphocytes

In the preliminary experiments, splenic Th cells-related dys-regulations were similar between TAA-V mice and BDL-V mice. Accordingly, only the splenic T lymphocytes of TAA-V mice were used to explore the acute effects of propranolol on Th cells. Splenocyte suspensions were prepared by cutting the tissue and then filtering through a 70-mm nylon membrane and washing with RPMI 1640 from GIBCO (Carlsbad, CA, USA). Red blood cells were lysed by adding 2 mL of room temperature lysis buffer (0.16 M NH4Cl, 10 mM KHCO_3_, and 0.13 mM EDTA) for 2 min, followed by one wash with HBSS/10% heat-inactivated fetal bovine serum (FBS). Each cell pellet was resuspended in HBSS, filtered and washed a final time in HBSS. Cells were counted, and nonviable cells were identified by uptake of trypan blue (Sigma, St. Louis, MO, USA) and samples were resuspended (2.5 × 10^6^ cells/mL) in supplemented RPMI medium (10% heat-inactivated FBS, 0.075% sodium bicarbonate, 10 mM HEPES buffer, 100 U/mL penicillin G, 100 mg/mL streptomycin sulfate, 1.5 mM *L*-glutamine, and 0.0035% 2-mercaptoethanol). The purity of sorted cell populations was typically 97%. Then, the EasySep™ Mouse Naïve CD4^+^ T Cell Isolation Kit (StemCell Technologies, Vancouver, Canada) was used to isolate CD4^+^ T cells from single-cell suspensions of splenocytes by negative selection to a purity of 95%, which was verified by flow cytometry. The isolated cells exhibited > 95% viability, which was confirmed by trypan blue dye exclusion. Unwanted cells were targeted for removal with biotinylated antibodies directed against non-T cells and streptavidin-coated magnetic particles. Labeled cells were separated using an EasySep™ magnet without using columns (manufacturer, city, country). The desired cells were transferred into a new tube. All the culture media were supplemented with 10% fetal calf serum (BI), penicillin, streptomycin, glutamine, and 2-mercaptoethanol (Life technology, Foster City, CA, USA), and the CD4+ T cells (representing T-helper cells) were cultured at 37 °C in 5 % CO_2_. Then, the *mRNA* levels of ADRB1, ADRB2, CD68, F4/80, iNOS, IL-10, TGFβ1, and Foxp3 in cell lysates of splenic Th cells.

### 2.7. Effects of Chronic Propranolol Treatment on the Basal Proportion of the Th Cell Subsets

Two-color flow cytometry was performed using a fluorescence activated cell sorter (FACS) Caliber Flow Cytometer (BD Biosciences, San Diego, CA, USA). The cell surface monoclonal antibodies utilized, included CD25, CD45RA, CD27, ADRB1, ADRB2, and intracellular monoclonal antibodies against forkhead box P3 (Fox P3). The proportion of naïve Th cells, effector memory Th cells and regulatory Th cells were measured by flow cytometry-based sorting of CD45RA^+^ CD27^+^ (% of naïve Th cells), CD45RA^−^ CD27^+^ plus CD45RA^+^ CD27^−^ plus CD45RA^−^ CD27^−^ (% of effector memory Th cells), and CD25^+^ FoxP3^+^ (% of regulatory T cells) among the CD4^+^ Th cells. Additionally, the percentage of ADB1^+^ ADB2^+^, proliferative (Ki-67^+^), apoptotic (Annexin-V^+^ PI^−^), and activated (HLADR^+^) cells among CD4^+^ Th cells was assessed. Isotype control antibodies were used to determine the level of background staining and helped to set up a specific signal threshold. Stained cells were analyzed using a FACSCalibur flow cytometer (BD Bioscience, Franklin Lakes, NJ, USA) and FlowJo software 7.6.1 (Tristar, Culver City, CA, USA).

### 2.8. Acute Effects of Propranolol on the Proliferation and Apoptosis of Splenic Lymphocytes

After activation with cell stimulation cocktail (500×) containing PMA (phorbol 12-myristate 13-acetate), ionomycin, Brefeldin A, and monensin for 4 days, splenic lymphocytes (200 µL, 1 × 10^6^ cells) from sham-V and TAA-V mice were incubated with buffer, propranolol (1 μM), propranolol + dobutamine (β1-agonist, 1 µM) or propranolol + salbutamol (β2-gonist, 1 µM) for 5 h. Then, apoptotic cells were assessed by TUNEL staining, and cell proliferation was assessed using the cell-counting kit, CCK-8 (Sigma-Aldrich, St. Louis, MO, USA), which measured the metabolic activity of living cells. For cell proliferation, 20 µL CCK-8 was added to each well, 4 h before the end of incubation. The absorbance was measured using Multiskan Spectrum from BioTek Co., Ltd., Winooski, VA, USA) at 450 nm. The proliferation index (%) was calculated using the following formula: OD of cells in wells of stimulation cocktail-incubated group/OD of cells in control wells. Cell lysates were harvested and analyzed for mRNA levels of ADRB1, ADRB2, caspase-3, and Ki-67 using the corresponding primers (Table 1).

### 2.9. Acute Effects of Propranolol on the Differentiation of Splenic Th Lymphocytes

For differentiation experiments of Th1, Th2, Treg conditions were created by coating the wells with anti-CD3 and anti-CD28 at 1 µg/mL in 1 mL of total volume (24-well plate) for 1 h at 37 °C. Then, each well was washed with 1 mL of sterile PBS (with no added FBS) and cultured with post-stimulated Th cells/well (1 × 10^6^) in 1.0 mL RPMI. To test the influence of β-adrenergic receptor on its differentiation, Th cells were pre-treated with propranolol (Pro, 1 μM), propranolol + dobutamine (ADRB1-agonist, Dob, 1 μM), or propranolol + salbutamol (ADRB2-agonist, Salb, 1 μM). Then, the cultured media were supplemented with corresponding reagents as specified below to induce the differentiation of post-stimulated Th cells: Th1: IL-2 (30 U/mL), IL-12 (20 ng/mL) and anti-IL-4 (10 μg/mL); Th2: IL-2 (30 U/mL), IL-4 (10 ng/mL), anti-IFN-γ (5000 ng/mL), and soluble mouse CD28 antibody (2 μg/mL, addition of soluble anti-CD28 into the plate-bound anti-CD28 has been found to enhance Th2 cytokine production); Treg: TGFβ (15 ng/mL), IL-2 (30 U/mL), anti-IFN-γ (5000 ng/mL) and anti-IL-4 (10 μg/mL) mAb.

Using the aforementioned flow cytometry-based intracellular cytokine staining analysis, the percentage of Th1-condition-differentiated IFN-γ producing Th (Th1) cells, Th2 condition-differentiated IL-10 producing Th (Th2) cells, and Treg condition-differentiated IL-35 producing Th (Treg) cells among fixed amount of the initially seeded splenic Th cells (1 × 10^6^) of either sham-V or TAA-V mice after acute pre-treatment with either propranolol, propranolol + dobutamine, or propranolol + salbutamol was calculated. Percentage of IFNγ^+^ IL-2^+^, IL-10^+^ IL-2^+^, and IL-35^+^ IL-2^+^ cells were calculated by flow cytometry under corresponding Th1/Th2/Treg differentiating conditions. Cell lysates were harvested and analyzed for protein and mRNA levels of T-bet (55 kD; from Santa Cruz Biotechnology, Dallas, TX, USA), GATA3 (50 kD; from Abcam, Cambridge, UK), ADRB1, ADRB2, Foxop3, PKA, p38MAPK and cAMP with corresponding antibodies and primers (Table 1).

### 2.10. Statistical Analysis

One-way analysis (ANOVA) was used to compare the different groups as appropriate. *p* < 0.05 was considered statistically significant.

## 3. Results

In comparison with sham mice, cirrhotic (BDL-V and TAA-V) mice were characterized by grossly observing cirrhotic livers, ascites, splenomegaly (increased spleen-to-BW ratio), higher plasma norepinephrine levels, and high incidence of systemic infection (Table 2 and Table 3). Higher incidence of infection in blood, intestines, pleural fluid, and liver was noted in cirrhotic mice than in sham mice. Interestingly, the incidence of aerobic bacterial infection in blood and intestine were higher than that in anaerobic bacterial infection (Table 3). In Table 2 and Table 3, the increased plasma norepinephrine levels, splenomegaly, and systemic infection were suppressed by chronic propranolol treatment in cirrhotic mice.

### 3.1. Chronic Propranolol Treatment Corrected Lymphopenia, Systemic Inflammation, and Bacterial Translocation in Cirrhotic Mice

Cirrhotic mice had the typical abnormalities of leucopenia, lymphopenia, thrombocytopenia, hypoalbuminemia, abnormal liver function, high serum C-reactive protein (CRP), and high lipopolysaccharide binding protein (LBP) levels (Table 1). In comparison with sham mice, significantly increased circulating Th1 (INF-γ and TNF-α) and Treg (IL-35) cytokines and decreased circulating Th2 (IL-10) cytokines were reported in cirrhotic mice, which were reversed by chronic propranolol treatment (Figure 1A–C).

### 3.2. Over-Expression of ADRB1 and ADRB2 Protein Was Found in Spleen and Splenic T Lymphocytes of Cirrhotic Mice

Among the tissues (liver, intestine, MLN, spleen) involved in the immune dysfunction in cirrhosis, simultaneous over-expression of ADRB1 and ADRB2 protein levels was only noted in the spleens of cirrhotic mice (Figure 2A). Basically, the number of CD3^+^ cells was lower in spleens of cirrhotic mice than those in sham mice. Particularly, a higher number of ADRB1^+^ CD3^+^ and ADRB2^+^ CD3^+^ cells was noted among CD3^+^ cells (T cells) from cirrhotic mice than those among sham mice (Figure 2B,C).

### 3.3. Chronic Propranolol Treatment Suppressed Abnormal Apoptotic Activity and Restored Naíve T and Effector Memory T Cells in Spleens of Cirrhotic Mice

Splenic white pulp size is regarded as a representative marker of proliferation or apoptotic activity of T cells. In our study, the white pulp area was significantly increased in cirrhotic mice and was rendered normal after propranolol treatment (Figure 3A). A significant increase in apoptotic (% of TUNEL^+^ area in splenic white pulp) activity was particularly observed in spleen of cirrhotic mice, which were inhibited by propranolol treatment (Figure 3B,C). Nonetheless, the slightly increased expression of proliferative protein, Ki-67, in the spleen of cirrhotic mice was not modified by propranolol treatment. About the splenic Th cells subsets of cirrhotic mice, the significantly decrease in naïve and effector T cells was associated with increased regulatory T cells subset and corresponding Treg cytokines (TGFβ1 and IL-10), which could in apoptosis of naïve and effect T cells. Significantly, the propranolol-related suppression of Treg subsets and cytokines was accompanied by the normalization of naïve and effector T cell subsets in spleen of cirrhotic mice (Figure 3D).

Interestingly, the above-mentioned abnormalities were accompanied by an increased percentage of ADRB1^+^ ADRB2^+^, proliferative (PCNA^+^ Ki-67^+^) and apoptotic (annexin-V^+^ PI^−^) Th cells, and decreased percentage of activated (CD69^+^ HLA-DR^+^) Th cells in spleen of cirrhotic mice, which were normalized by chronic propranolol treatment (Figure 4). Additionally, increased mRNA and protein levels of splenic ADRB1/ADRB2, inflammatory (CD68, F4/80 for macrophage), Th1 (INF-γ), and Treg (TGFβ1; FoxP3) cytokines were noted in cirrhotic mice (Figure 4C and Figure 5A) and their levels reduced after propranolol treatment, whereas Th2 cytokines (IL-4, IL-10) levels were reduced in cirrhotic mice and became improved after propranolol treatment. Taken together, the results indicated that chronic propranolol treatment normalized the percentage of naïve and effect T cells subsets and activation of Th cell through the suppression of Treg subsets and Treg inhibitory cytokines release in the spleens of cirrhotic mice.

### 3.4. High Levels of Peripheral and Splenic LBP Were Associated with High Splenic Levels of ADRB Protein and Th1/Treg Cytokines in Cirrhotic Mice

In comparison with the sham-V group, significant increase in splenic ADRB1 and ADRB2 protein levels were accompanied by increased splenic Th1/Treg cytokines (Figure 5A–C). Meanwhile, the corresponding imbalance in splenic Th1, Th2, and Treg cytokines was parallel to their similar trend in circulation of cirrhotic mice (Figure 1 and Figure 5B,C). Next, the cirrhotic mice (*n* = 28) were divided into “low” (< third percentile of increased percentage of ADRB1 plus ADRB2 protein level compared to sham-V group) and “high” (> third percentile) splenic ADRB groups. The separation of “low” and “high” splenic LBP groups was same as that of ADRB using the cut-off point as third percentile in comparison with “low” splenic ADRB group. Notably, higher splenic Th1 (INF-γ) and Treg (IL-35) cytokines and lower splenic Th2 (IL-10) cytokine were observed among “high” splenic ADRB group of cirrhotic mice (Figure 5D–F). Furthermore, increased splenic LBP levels were accompanied by the above-mentioned dominant changes in Th1/Treg cytokines of cirrhotic mice (Figure 5A–G). Interestingly, the magnitude of increase in splenic ADRB protein level was higher in the “high” splenic LBP group than that in the “low” splenic LBP group (Figure 5H). the mean high splenic LBP level was observed in the “high” ADRB group more than in the “low” ADRB group (Figure 5I). Effectively, chronic propranolol treatment inhibited the increase in splenic ADRB, Treg cytokines, and LBP levels in cirrhotic mice (Figure 5A–C,G).

### 3.5. Acute Propranolol Incubation Inhibits Apoptotic and Treg Markers of Splenic Th Cells from Cirrhotic Mice

In comparison with the sham-V group, significantly increased apoptosis [percentage TUNEL^+^ and CD3^+^ casp-3^+^ and cell lysate capase-3 *mRNA* levels] and slightly increased proliferation (CCK-assessed index and cell lysate’s Ki-67 *mRNA* levels) were observed in splenic Th cells isolated from the TAA-V group (Figure 6). The up-regulation of ADRB1 and ADRB2 *mRNA* levels were accompanied by alteration of its down-stream signals [increased cAMP and decreased *p38MAPK mRNA* levels] in lysates of Th cells isolated from the TAA-V group (Figure 7A–C). Additionally, increased T-bet (Th1) and Foxp3 (Treg) protein levels were associated with a decrease in *GATA* (Th2) *mRNA* levels in Th cell lysates isolated from the TAA-V group (Figure 7A–C).

A similar degree of inhibition of ADRB1 and ADRB2 levels indicated the adequacy of the concentration of the acute propranolol incubation (Figure 7A). Significantly, acute propranolol treatment suppressed the apoptotic markers, reduced the Treg marker, increased the Th2 marker; however, it did not modify the proliferative marker of the TAA-V group’ splenic Th cells and its cell lysates (Figure 5 and Figure 6). Propranolol incubation-related suppression of apoptosis and corresponding signals in the TAA-V group splenic Th cells were equally counteracted by co-incubation with β2 agonist salbutamol or β1 agonist dobutamine (Figure 6).

### 3.6. Propranolol Inhibited Treg-Conditioned Differentiation and Restored Th2-Conditioned Differentiation of Th Cells Isolated from Cirrhotic Mice

In vitro experiments revealed that, in comparison with the sham-V group, splenic Th cell from cirrhotic mice were characterized by increased Treg-conditioned differentiation and decreased Th2-conditioned differentiation (Figure 7D). Significantly, acute propranolol incubation reduced Treg-conditioned differentiation and enhanced Th2-conditioned differentiation of splenic Th cell from cirrhotic mice. Notably, acute concomitant ADRB2 agonist (salbutamol) incubation rather than ADRB1 agonist (dobutamine incubation blocked the abovementioned propranolol-related effects on Treg and Th2-conditioned differentiation of cirrhotic splenic Th cells (Figure 7D). In fact, the Th1-conditioned differentiation of splenic Th cells was notably modified by acute propranolol pre-treatment.

## 4. Discussion

The spleen is important for protecting individual from infection and inflammation through the regulation of the apoptosis, proliferation, activation, and differentiation of Th cells. Bacterial translocation, defined as high circulating LBP level in cirrhosis, is the main contributor to the increased proliferation and apoptosis, and decreased activation of naíve and effector memory Th cells. Total and T-helper-cell lymphopenia is a common finding in cirrhotic patients, which is usually caused by splenic sequestration [5,22]. It had been reported that chronic propranolol treatment can increase the number of circulating T cell [23]. Pre-surgery administration of propranolol can restore the depletion of CD4^+^ T-helper cell after surgery [24]. Bacterial translocation-related stimulation of Th-cell apoptosis is the major pathogenic factor for the lymphopenia-related immune dysfunction in cirrhosis. In our study, with an improvement of portal hypertension and splenomegaly by chronic propranolol treatment in cirrhotic mice, the splenic T lymphopenia, bacterial translocation, systemic infection, and inflammation were suppressed. Particularly, the elevation of splenic LBP as well as up-regulation of splenic ADRB1 and ADRB2 expression indicated the involvement of the spleen in the pathogenesis of cirrhosis-associated immune dysfunction. It had been reported that the intestinal immune dysregulation and mesenteric Th1 polarization driven bacterial translocation in cirrhotic rats [7,25]. Especially, the expansion of intestinal epithelial and lamina propria lymphocytes T cells and effector T cells subsets precede the development of bacteria translocation in cirrhosis [25]. Taken together, it is possible that the expansion of intestinal total T/effector T cells induce intestinal inflammation, promote BT, activate and exhausted over-activated splenic T cells, and finally immune dysfunction in advanced cirrhosis.

It had been reported that increased plasma norepinephrine levels will suppress the activation and function of splenic T cells [26]. Both our—and other previous studies—reported that higher plasma levels of norepinephrine were observed in cirrhotic animals than those in normal controls [12,27,28]. It had been suggested that increased plasma level of norepinephrine reflect the sympathetic nerve system (SNS) hyperactivity [29]. In our study, higher plasma norepinephrine level is accompanied by the over-activated splenic SNS and over-expressed splenic ADBRs in cirrhotic mice. Significantly, chronic administration of propranolol in our study decreased the plasma levels of norepinephrine in cirrhotic mice. These results indicated that the beneficial effects of chronic propranolol treatment in cirrhotic mice is contributed the suppression of splenic SNS and down-regulated splenic ADBR expression in cirrhotic mice.

It had been reported that chronic endotoxemia over-activates splenic SNS and induces T lymphopenia-related immune dysfunction [30,31]. In cirrhotic rats with ascites, splanchnic sympathectomy can effectively reduce bacterial translocation [13]. A β-adrenergic effect has been reported to be involved in the modulation of T lymphocytes homeostasis in rat spleen [32]. In our study, the inhibition of splenic ADRB1 and ADRB2 expressions by chronic propranolol treatment was accompanied by redistribution of T cell subsets and the corresponding cytokines in circulation and in the spleen of cirrhotic mice. Overall, chronic propranolol treatment decreased BT and corrected cirrhosis-associated immune dysfunction by normalizing naïve cell and effector memory Th cells, restored Th2-condiotioned differentiation and decreased Treg-conditioned differentiation in the spleen of cirrhotic mice.

Many ADRB binding sites had been reported in Th cells. Studies, in which ADRB1 was blocked, provided evidence for its immunomodulatory role in host immunity [33]. ADRB1 antagonist can abolish cold restraint-related decreases in bacterial clearance. Excessive simulation of ADRB1 is responsible a decrease in LPS-induced TNF-α response in mice [34]. Significantly, propranolol normalizes the socially disruptive stress-related suppression of the LPS-stimulated TNF-α production by murine immune cells [20]. Under stress, T lymphopenia is caused by stress hormone-induced apoptosis of freshly release circulating splenic lymphocytes, which display a high ADRB2 density [35]. In our study, the propranolol treatment suppressed the splenic TNF-α level, preserved activation ability, and decreased apoptotic activity in the spleens and splenocytes of cirrhotic mice. Specifically, in vitro experiments revealed that both ADRB1 and ADRB2 mediated the propranolol-related anti-apoptotic and activating effects in Th cells. In addition, ADRB2 mediates the propranolol-related effects, including the promotion of the Th2-conditioned differentiation and inhibition of the Treg-conditioned differentiation of Th cells. Impaired monocyte function, including defects in chemotaxis, phagocytosis and killing activity, as well as a decrease in the production of lysosomal enzymes, are well-known components of cirrhosis-associated immune dysfunction [36]. On mononuclear cells (MNC) from peripheral blood, although the density or affinity of ADRB2 binding sites are normal, however, the number of binding sites per cell was significantly lower in patients with severe ascites than in patients with mild to moderate or no ascites [37]. Therefore, the impacts of chronic propranolol treatment on the expression of ADRB2 on MNC from peripheral blood need to evaluate in future studies.

Splenic white pulp (WP) is a T cell rich area that generates immune responses to protect the body against blood-borne infections. Generally, the pool of naïve Th cells depends on the proliferation and apoptosis of these cells and their differentiation into effector memory Th cells [38]. In splenic WP, naïve T cells undergo activation and differentiate into effector memory T cells, which can penetrate inflamed tissues to clear them from pathogens [39]. In our study, the normalization of the WP areas of cirrhotic spleen was accompanied by the restoration of the percentage of effector memory T cells among splenic Th cells.

In the cirrhotic mice of this study, the decrease in the number of peripheral lymphocyte is accompanied by increased circulating Treg cytokines, which could induce the apoptosis of T cells. The differentiation of naíve T-helper cells towards Th2 or Treg cells is regulated by T cell-expressed transcription factors, GATA-binding protein-3 (GATA-3) and Foxp3 [40]. Therefore, in the current study, the restoration of Th2-conditioned differentiation and suppression of Treg-conditioned differentiation of splenic Th cells by acute propranolol treatment was associated with increased GATA-3 and decreased Foxp3 levels in cell lysates of Th cells from cirrhotic mice.

Under normal circumstances, Th17 and Treg cells maintain a dynamic balance and result in the induction of immune responses of appropriate intensity, which is conducive to the maintenance of a stable immune state in the body. It had been reported that the degree of disequilibrium between Th17 and Treg level is correlated with the severity of systemic inflammation in mice with post-cerebral ischemia and collagen-induced arthritis [41,42]. Treg/Th17 imbalance is involved in the pathogenesis of liver cirrhosis and predicting the decompensation of liver cirrhosis [43]. T lymphopenia and immune dysfunction usually occur in patients with advanced cirrhosis, whose are characterized by abnormal hepatic neo-angiogenesis, hepatic ischemia, and systemic inflammation [5,7,21,44]. In the current study, we mainly explore the effects of chronic propranolol treatment on the abnormal changes in the Th1, Th2, Treg cytokines in mice with cirrhosis-related immune dysfunction. Therefore, in future studies, it is necessary to explore the effects of chronic propranolol treatment on imbalance between Th17 and Treg cell on the cirrhosis-associated immune dysfunction.

Stimulation with an ADRB2 agonist promotes the conversion of naïve T cells to Treg cells, which express ADRB2 [45]. In our study, we reasonably, to observe, that the up-regulation of ADRB2 expressions on cirrhotic splenic Th cells was associated with an increased percentage of Treg cells in circulation and in the spleen. Tregs can release immunosuppressive cytokines including TGF-β, IL-10 and IL-35 [46]. Plasma levels of TGF-β1 are significantly correlated with hepatic TGF-β1 content indicating that plasma levels can reflect tissue expression [47]. Moreover, increased *mRNA* levels of TGF-β1 are positively correlated with the degree of immune cell infiltration in the livers of chronic liver disease patients [48]. In the presence of the exogenous TGF-β, naíve Th cell up-regulates the expression of the Tregs’ marker Foxp3 and are converted to immunosuppressive role of Treg cells [48,49,50,51]. In our study, the amelioration of cirrhosis-associated immune dysfunction by propranolol treatment was accompanied by a decrease in the frequency of Treg cells and corresponding TGFβ1, IL-10 and IL-35 levels in circulation and in the spleen.

The immunosuppressive effects of Treg cells are correlated with the levels of its surrogate marker Foxp3 [52,53]. In the cirrhotic mice in our study, it is reasonable to infer that chronic propranolol treatment suppress the differentiation of naíve Th cells into Tregs by down-regulating TGFβ, Foxo3, IL-10, and IL-35 expressions in spleen.

In the current study, the chronic effects of propranolol treatment on the cirrhosis-related Th-cell depletion, systemic inflammation, BT, and infection were explored in BDL and TAA rats. Nonetheless, to dissect differences in the etiologies of observed changes in T cell phenotype/numbers, additional studies with mice after chronic isoproterenol-application (no pharmacological model of increased SNS-activity) and PVL mice with portal hypertension and splenomegaly (healthy liver and less SNS-activity) are needed future studies.

In vitro studies reported that acute incubation of propranolol promote apoptosis and inhibit proliferation of cultured endothelial cells (ECs) [54,55]. In cirrhosis, it had been reported that splanchnic, portal, hepatic and pulmonary angiogenesis are mainly contributed to the portal hypertension (PH) and various clinical complications [44]. Systemic angiogenesis induces increased portal inflow and portosystemic collaterals as well as lethal complications of PH such as gastroesophageal variceal hemorrhage (GEV). Propranolol remains the mainstay of pharmacologic treatment for PH and GEV. In cirrhotic portal hypertensive rats, chronic propranolol improves portal hypertension by decreasing in the extent of portal-systemic shunting, splanchnic/hepatic angiogenesis, and liver fibrosis [21]. Thus, in future studies, in addition to the beneficial effects of chronic propranolol treatment on immune dysfunction of cirrhosis in the current study, the beneficial effects of chronic propranolol on the parallelly existed cirrhosis-related angiogenesis should be explored. Figure 8 show the graphical summary of the effects of chronic propranolol treatment the on the regulation, modulation, and changes that are associated with cirrhosis in terms of SNS-activity and T-lymphocytes.

## 5. Conclusions

In conclusion, this study suggests that the over-expression of splenic ADRB1 and ADRB2 is significantly involved in the pathogenesis and mechanism of cirrhosis-associated immunological dysfunction. In addition to its effects on hemodynamics, through down-regulation of splenic ADRB1 and ADRB2 expressions, the non-selective ADRB antagonist propranolol effectively normalized the homeostasis and function of T cells subsets in the microenvironment of cirrhotic spleen. Notably, the above-mentioned benefits on the immune system are associated with a decrease in incidence of systemic inflammation, systemic infection; thus, leading to an improvement in the severity of cirrhosis-associated immune dysfunction. The results of this study open a new window towards therapeutic effects of propranolol in cirrhosis.

## Figures and Tables

**Figure 1 cells-09-00604-f001:**
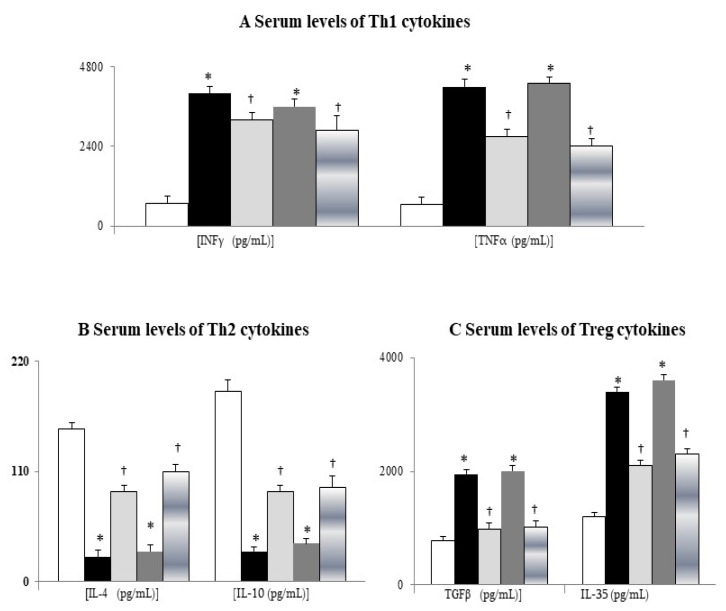
Serum T-helper 1 (Th1)/Treg cytokines were increased and serum Th2 cytokines were decreased in cirrhotic mice. Serum levels of Th1 (**A**), Th2 (**B**), and Treg (**C**) cytokines. * *p* < 0.05 vs. sham-V group; ^†^
*p* < 0.05 vs. BDL-V/TAA-V group.

**Figure 2 cells-09-00604-f002:**
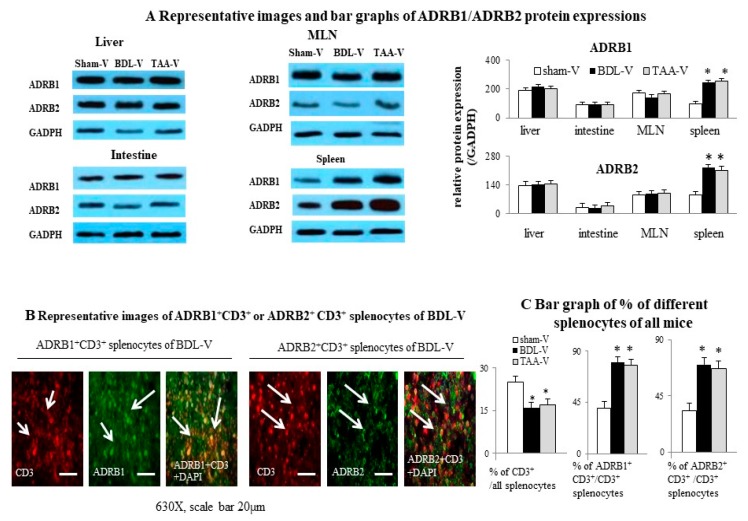
Over-expressions of β1 and β2 adrenergic receptors (ADRB1 and ADRB2) proteins were noted in spleen and splenic T lymphocytes of cirrhotic mice. (**A**) Representative images and bar graphs of ADRB1/ADRB2 protein expressions in liver, intestines, mesenteric lymph node (MLN), and spleen in sham-V and cirrhotic (BDL-V/TAA-V) mice; (**B**) Representative immunofluorescence images of ADRB1^+^ CD3^+^ or ADRB2^+^ CD3^+^ splenocytes of BDL-V mice. White arrows represent positive of CD3, ADRB1, ADRB2, CD3 and ADRB1 or CD3 and ADRB2; (**C**) Bar graphs of percentages of CD3^+^, ADRB1^+^ CD3^+^ and ADRB2^+^ CD3^+^ splenocytes of sham-V or cirrhotic (BDL-V/TAA-V) mice. * *p* < 0.05 vs. sham-V group.

**Figure 3 cells-09-00604-f003:**
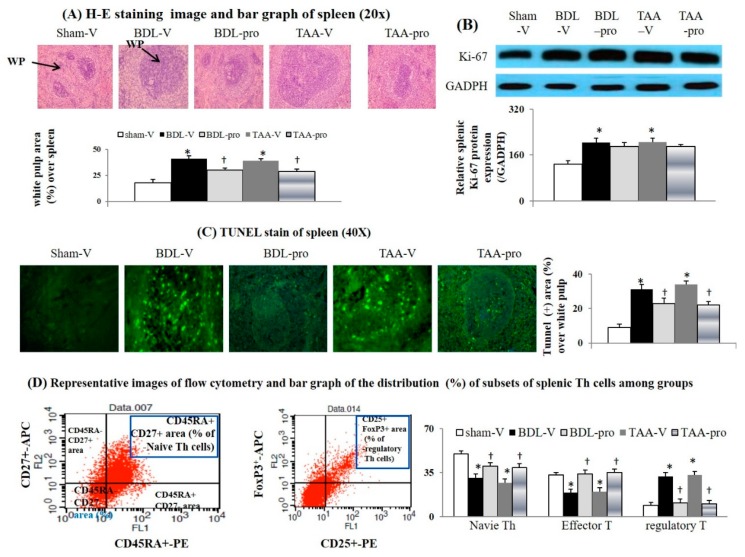
Chronic propranolol treatment suppressed the splenic white pulp (WP) hyperplasia, inhibited abnormal proliferative/apoptotic activities of spleen and modified distribution of the Th cell subsets in cirrhotic mice. (**A**) Representative images and bar graphs of H-E stain of splenic WP area; (**B**) Splenic Ki-67 protein expression; (**C**) Representative immunofluorescence images and bar graphs for the splenic TUNEL (+) area; (**D**) The distribution of splenic naïve T (% of CD45RA^+^ CD27^+^) cell, effector memory T (% of CD45RA^−^ CD27^+^ plus CD45RA^+^ CD27^−^ plus CD45RA^−^ CD27) cells and regulatory T (% of CD25^+^ FoxP3^+^) cells among CD4^+^ Th cells. * *p* < 0.05 vs. sham-V group; ^†^
*p* < 0.05 vs. BDL-V/TAA-V group.

**Figure 4 cells-09-00604-f004:**
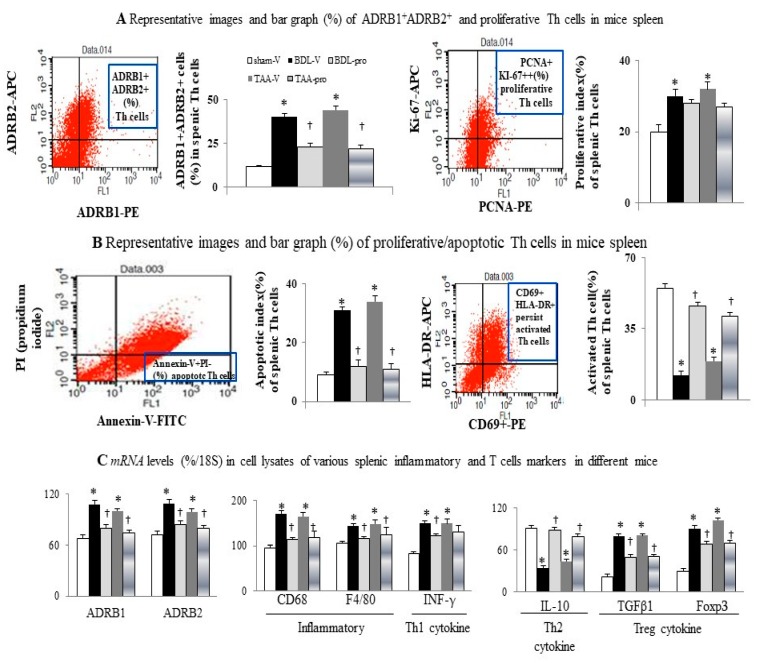
Increased percentage of ADRB1^+^ ADRB2^+^ cells was associated with increased percentage of proliferative/apoptotic Th cells and decreased activated Th cells in spleen of cirrhotic mice. The percentage distribution of (**A**) ADRB1^+^ ADRB2^+^ and proliferative (Ki-67^+^ PCNA^+^) Th cells among groups, and (**B**) apoptotic [Annexin-V^+^ PI^−^], and activated [HLA-DR^+^ CD69^+^] Th cells among groups; (**C**) mRNAs level (%/18S) in cell lysates of various splenic inflammatory and T cells markers of Th cells from different groups. * *p* < 0.05 vs. sham-V group; ^†^
*p* < 0.05 vs. BDL-V/TAA-V group.

**Figure 5 cells-09-00604-f005:**
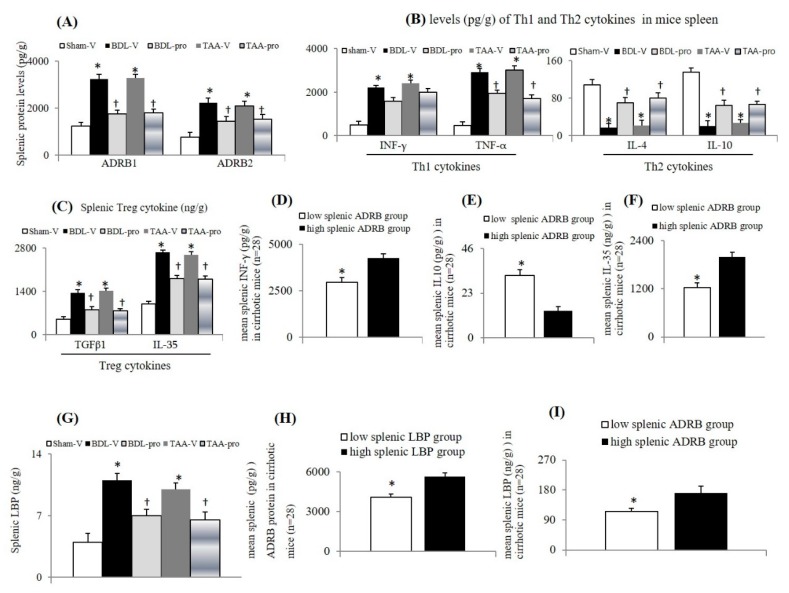
High tissue levels of ADRB1/ADRB2 proteins were associated with high levels of Th1/Treg cytokines and LBP in the spleens of cirrhotic mice. (**A**–**C**) levels of ADRB1/ADRB2 proteins and of Th1/Th2/Treg cytokines; (**D**–**F**) Mean levels of INF-γ, IL-10 and IL-35 in cirrhotic mice of “high” ADRB and “low” ADRB groups. (**G**) Splenic LBP level between groups; (**H**) LBP levels in cirrhotic mice of “high” ADRB group and “low” ADRB group. (**I**) ADRB protein Levels between “high LBP” and “low LBP” groups. In cirrhotic mice, “high” ADRB or LBP groups were defined by high percentages (> third percentile, cut-off point is 393 for ADRB protein levels, and 1057 for splenic LBP levels) of summative increased splenic ADRB1 and ADRB2 protein or LBP levels compared to sham-V mice. * *p* < 0.05 vs. sham-V/high ADRB or LBP groups; ^†^
*p* < 0.05 vs. BDL-V/TAA-V group; LBP: lipopolysaccharide binding protein.

**Figure 6 cells-09-00604-f006:**
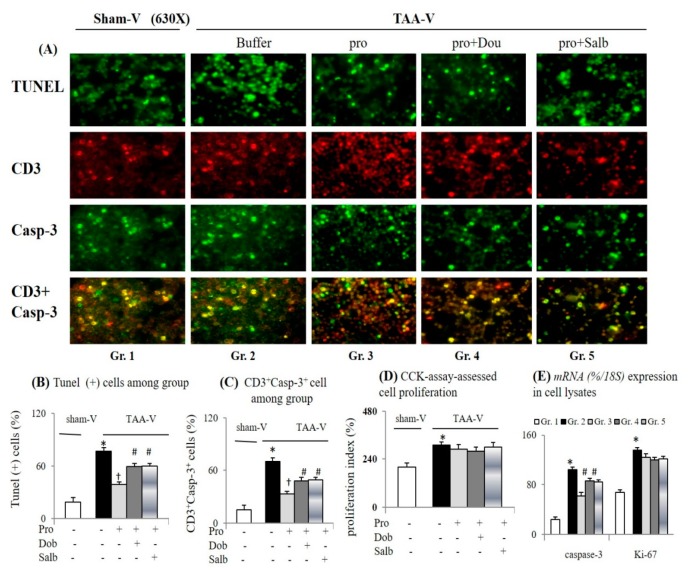
Acute propranolol incubation inhibited apoptosis of splenic Th cells isolated from cirrhotic mice. (**A**–**C**) Representative images and bar graphs of TUNEL/CD3 plus Casp-3 immunofluorescence (IF) staining of naïve Th cells isolated from sham-V and TAA-V mice with acute incubation of propranolol (pro), pro + dobutamine (dob) or pro + salbutamol (sal); (**D**) cell-counting kit (CCK)-assessed proliferation index; (**E**) mRNAs (%/18S) levels in cell lysates; Gr. 1: buffer-treated sham-V group cells; Gr. 2: buffer-treated TAA-V group’ cells; Gr. 3: pro-treated TAA-V group’ cells; Gr. 4: pro + dob-treated TAA-V group’ cells; Gr. 5: pro + sal-treated TAA-V group’ cells. * *p* < 0.05 vs. sham-V group; ^†^
*p* < 0.05 vs. TAA-V group; ^#^
*p* < 0.05 vs. TAA-pro group.

**Figure 7 cells-09-00604-f007:**
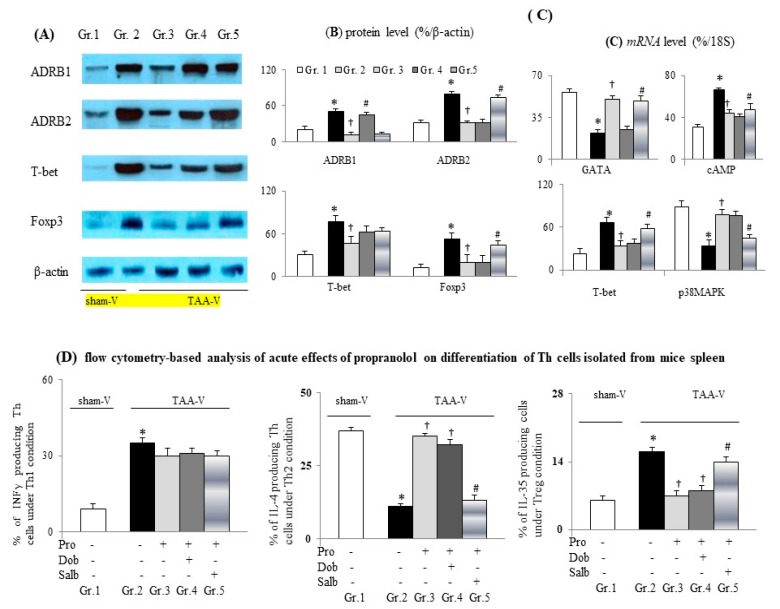
Acute propranolol incubation inhibited Treg-conditioned differentiation and restored Th2-conditioned differentiation of Th cells isolated from the spleens of cirrhotic mice. (**A**,**B**) Representative images and bar graphs of proteins and *mRNAs* (%/18S); (**C**) Expressions in cell lysates of Th cells isolated from spleen of sham-V or TAA-V mice; (**D**) Flow cytometry-based intracellular cytokine staining for percentages of Th1/Th2/Treg-conditioned differentiation among groups. * *p* < 0.05 vs. sham-V group; ^†^
*p* < 0.05 vs. TAA-V group.

**Figure 8 cells-09-00604-f008:**
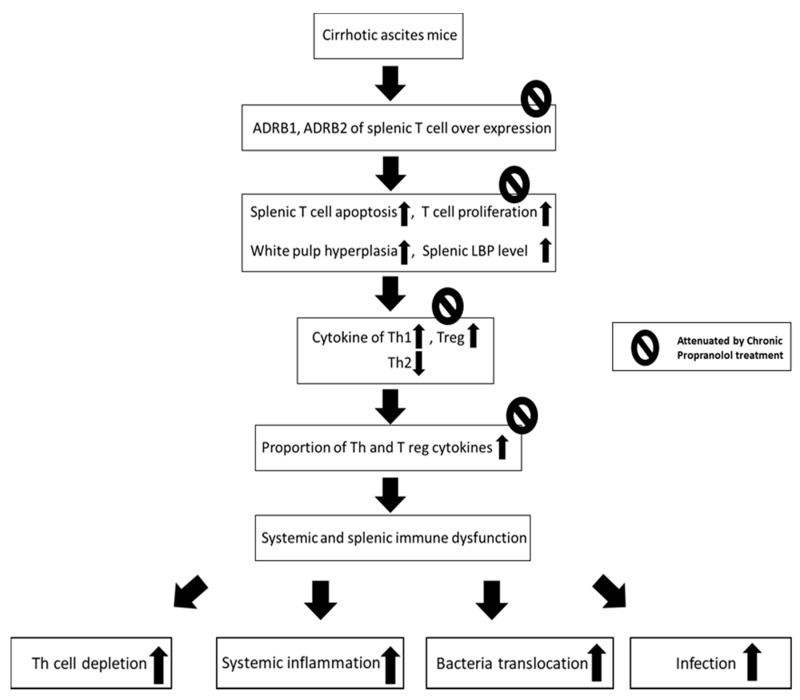
Graphical summary of the effects of chronic propranolol treatment the on the regulation, modulation, and changes that are associated with cirrhosis in terms of sympathetic nervous system (SNS)-activity and T-lymphocytes. SNS, sympathetic nervous system; ADRB1/ADRB2, β1 and β2 adrenergic receptors; Th cell, helper T cell; Treg, regulatory T cell.

**Table 1 cells-09-00604-t001:** Primer sequences used for various genes expression analysis by real-time qPCR.

Name of Gene	Sequence of Sense Primer (5′-3′)	Sequence of Anti-Sense Primer (3′-5′)
18S	ACGGAAGGGCACCACCAGGA	CACCACCACCCACGGAATCG
ADRB1	CGTCGCCCTTTCGCTACCAG	CCGCCACCAGTGCAGTGCTGAGGAT
ADRB2	TGCGTGATTGCAGTGGATCGCTAT	CTATCTTCTGCAGCTGCCTTTTGG
CD68	GCTACATGGCGGTGGAGTACAA	ATGATGAGAGGCAGCAAGATGG
F4/80	CAAGACTGACAACCAGACG	ACAGAAGCAGAGATTATGACC
IFN-γ	TGA ACG CTA CAC ACT GCA TCT TGG	CGA CTC CTT TTC CGC TTC CTG AG
IL-10	ACAGCCGGGAAGACAATAACT	ACACCCAGGAAAGACAGCA
TGFβ1	CCTGCAAGACCATCGACATG	TGTTGTACAAAGCGAGCACC
T-bet	CGG CTG CAT ATC GTT GAG GT	GTC CCC ATT GGC ATT CCT C
GATA-3	TGTCTGCAGCCAGGAGAGC	ATGCATCAAACAACTGTGGCCA
Foxp3	ACACCCAGGAAAGACAGCA	ACACCCAGGAAAGACAGCA
Caspase-3	GGTATTGAGACAGACAGTGG	CATGGGATCTGTTTCTTTGC
Ki-67	ATTTCAGTTCCGCCAATCC	GGCTTCCGTCTTCATACCTAAA
cAMP	AGAAATCACCCAGCAGGGCAAA	GTATGGGGACAGTGACCCTCAACC
p38MAPK	CGAAATGACCGGCTACGTGG	CACTTCATCGTAGGTCAGGC

ADRB1: β1-adrenergic receptor; ADRB2: β2-adrenergic receptor.

**Table 2 cells-09-00604-t002:** Basal characteristics of all mice.

	Sham (*n* = 4)	BDL-V (*n* = 7)	BDL-Pro (*n* = 7)	TAA-V (*n* = 7)	TAA-Pro (*n* = 7)
Presence of ascites (%)	0	6/7	5/7	4/7	3/7
Body weight (BW, gm)	29.7 ± 1.6	27.1 ± 2.4	28.3 ± 1.9	26.1 ± 1.1	27.5 ± 0.9
Liver weight (mg)	1.2 ± 0.2	2.51 ± 0.3 *	2.6 ± 0.1 *	2.32 ± 0.4 *	2.4 ± 0.1 *
Liver to BW ratio (mg/100g BW)	4.1 ± 0.6	9.2 ± 0.5 *	9.1 ± 0.3 *	8.9 ± 0.2 *	8.7 ± 0.1 *
Spleen weight (mg)	0.184 ± 0.01	0.27 ± 0.02 *	0.21 ± 0.03	0.24 ± 0.01 *	0.202 ± 0.01
Spleen to BW ratio (mg/100 g BW)	0.62 ± 0.01	0.99 ± 0.02 *	0.7 4± 0.01 ^#^	0.92 ± 0.01 *	0.73 ± 0.02 ^#^
Plasma norepinephrine level (pg/mL)	298 ± 64	549 ± 66 *	398 ± 77 ^#^	568 ± 95 *	398 ± 87 ^#^

** p* < 0.05 vs. sham-V, ^#^
*p* < 0.05 vs. bile duct ligation (BDL)-V/thioacetamide (TAA)-V mice.

**Table 3 cells-09-00604-t003:** The positive bacterial culture in cultured organs of all mice.

		Sham (*n* = 4)	BDL-V (*n* = 7)	BDL-Pro (*n* = 7)	TAA-V (*n* = 7)	TAA-Pro (*n* = 7)
Blood	aerobic bacteria	1/4(25%)	5/7(71% *^,^^δ^)	3/7(43% ^#^)	6/7(86% *^,^^δ^)	2/7 (29% ^#^)
	anaerobic bacteria	0/4(0%)	3/7(42% *)	1/7 (14% ^#^)	4/7(57% *)	1/7 (14% ^#^)
Ascites	aerobic bacteria	-	4/7(57% *^,^^δ^)	2/7 (29% ^#^)	5/7(71% *^,^^δ^)	1/7 (14% ^#^)
	anaerobic bacteria	-	5/7(71% *)	3/7 (43% ^#^)	3/7(43% *)	1/7 (14% ^#^)
Lung	aerobic bacteria	0/4(0%)	2/7(29% *)	2/7 (29%)	1/7(14% *)	0/7 (0%)
	anaerobic bacteria	0/4(0%)	1/7(14%)	1/7 (14%)	2/7(29%)	0/7 (0%)
Intestine	aerobic bacteria	0/4(0%)	6/7(86% *^,^^δ^)	2/7 (29% ^#^)	5/7(71% *^,^^δ^)	1/7 (14% ^#^)
	anaerobic bacteria	0/4(0%)	2/7(29% *)	1/7 (14%)	1/7(14% *)	0/7 (0%)
Pleural fluid	aerobic bacteria	-	2/7(29% *)	0/7 (0%)	3/7(43% *^,^^δ^)	1/7 (14% ^#^)
	anaerobic bacteria	-	2/7(29% *)	1/7 (14%)	2/7(29% *)	1/7 (14% ^#^)
Liver	aerobic bacteria	1/4(25%)	3/7(43% *)	1/7 (14% ^#^)	2/7(29% ^δ^)	1/7 (14% ^#^)
	anaerobic bacteria	1/4(25%)	2/7(29%)	1/7 (14%)	3/7(43% *)	1/7 (14% ^#^)

* *p* < 0.05 vs. sham-V, ^#^
*p* < 0.05 vs. BDL-V/TAA-V rats; ^δ^
*p* < 0.05 vs. anaerobic positive bacteria culture rates.

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
