# Peer review of "Propranolol Suppresses the T-Helper Cell Depletion-Related Immune Dysfunction in Cirrhotic Mice"

_cells, 2020, doi:10.3390/cells9030604_

Round 1

Reviewer 1 Report

Originality

This investigation is very timely, sufficiently original since mechanisms for CAID are ill defined and not addressed so far; Moreover, the clinical relevance is of potential huge impact.

Methods

Dose of propranolol 30mg/kg/day orally for 3 weeks is routine/ standard or well-chosen after dose-findings ?

Models used are BDL and TAA. However, no pharmacological model of increased SNS-activity, particularly since focusing on beta-adrenoreceptors a model of e.g. chronic isoproterenol-application would have been highly attractive. In addition, adding PVL-mice with portal hypertension and likewise splenomegaly but healthy liver and less SNS-activity would be helpful to dissect differences in etiology of observed changes in T-cell phenotyp/numbers.

Splenic T-lymphocytes but no other compartment/s are evaluated. In contrast, to ease comparison with data in the literature serum levels of Th-cells seem mandatory.

Can the authors present data on the levels of catecholamines (e.g. norepinephrine or as long-standing excessive SNS-activation NPY) ? in other words the expression levels of ADRs and associated changes do relate to the degree of SNS-activity ? although this reviewer is well aware that blood levels do not reflect organ SNS-drive.

Results
positive blood cultures in sham animals are not realistic and raise doubts on execution of experiments and in fact, reproducibility.

Interpretation

MNC from peripheral blood have been reported to present with down-regulation of beta-adrenoreceptors in ascitic cirrhotic patients (Gerbes A et al. Lancet 1986). Moreover, Munoz et al. reported an expansion of Th and Teffector cells (Hepatology 2019). The authors should at least discuss this controversy

Presentation:

Table 1 does not show what is metioned in text (page 6/18)
Fig. 1: symbols for groups not apparent for each subset of images
can the authors present a graphical summary on the regulation, modulation and changes associated with cirrhosis in terms of SNS-activity and T-lymphocytes ?

Author Response

Reply to Reviewer 1’s comments

Comment 1: Originality

This investigation is very timely, sufficiently original since mechanisms for CAID are ill defined and not addressed so far; Moreover, the clinical relevance is of potential huge impact.

-Thanks for giving us this opportunity to improve our manuscript.

Comment 2:  Methods: Dose of propranolol 30mg/kg/day orally for 3 weeks is routine/ standard or well-chosen after dose-findings?

Answer 2: Thanks for your importance comment about the dose of propranolol. As we reported in method section of text ”The dose of propranolol was based on the hemodynamic beneficial effects of in cirrhotic rats [Huang YT, Lin HC, Tsai JF, Hou MC, Hong CY (1997). Chronic administration of propranolol improves vascular contractile responsiveness in portal hypertensive rats. Eur. J. Clin. Invest. 27(7):550-555; D'Amico, M., Mejias, M., García-Pras, E., et al. (2012) Effects of the combined administration of propranolol plus sorafenib on portal hypertension in cirrhotic rats. Am. J. Physiol. Gastrointest. Liver. Physiol. 302,G1191-1198].

It had been reported that, in CBDL-cirrhotic rats, 2 week of propranolol treatment can effectively decrease portal pressure by reducing portal blood flow, decreasing superior mesenteric artery (SMA) blood flow, reducing SMA neovasculization, suppressing hepatic fibrosis, inhibiting hepatic neovascularization, and reducing portosystemic shunting [D'Amico, M., Mejias, M., García-Pras, E., et al. (2012) Effects of the combined administration of propranolol plus sorafenib on portal hypertension in cirrhotic rats. Am. J. Physiol. Gastrointest. Liver. Physiol. 302,G1191-1198]. In another study of portal vein ligated rats, 9 days of chronic propranolol (30 mg kg-1 day-1) treatment significantly alleviated the hyperdynamic state, including portal pressure, cardiac index and total peripheral resistance by improvement of arterial contractile reactivity of superior mesenteric artery (SMA) [Huang YT, Lin HC, Tsai JF, Hou MC, Hong CY (1997). Chronic administration of propranolol improves vascular contractile responsiveness in portal hypertensive rats. Eur. J. Clin. Invest. 27(7):550-555]. In “revsied” version, we had rewritten and checked the description about this important point [page 3, paragraph 2, line 4-10].

Comment 3: Models used are BDL and TAA. However, no pharmacological model of increased SNS-activity, particularly since focusing on beta-adrenoreceptors a model of e.g. chronic isoproterenol-application would have been highly attractive. In addition, adding PVL-mice with portal hypertension and likewise splenomegaly but healthy liver and less SNS-activity would be helpful to dissect differences in etiology of observed changes in T-cell phenotype/numbers.

Answer 3: Thanks for importance opinion about additional studies with mice after chronic isoproterenol-application [no pharmacological model of increased SNS-activity], PVL mice with portal hypertension and splenomegaly [healthy liver and less SNS-activity] in future studies to dissect differences in the etiologies of observed changes in T-cell phenotype/numbers. These importance future works had been included in the limitation of this study in the “discussion” section of “revised version [page 17, paragraph 4].

Comment 4: Splenic T-lymphocytes but no other compartment/s are evaluated. In contrast, to ease comparison with data in the literature serum levels of Th-cells seem mandatory.

Answer 4: Thanks for your importance comments about the importance for literature review of serum levels of Th-cells in cirrhosis. It had been reported that the number of peripheral total T cell and subset of T helper cells were depleted in cirrhotic patients [Lario, M., Muñoz, L., Ubeda, M., et al. (2013) Defective thymopoiesis and poor peripheral homeostatic replenishment of T-helper cells cause T-cell lymphopenia in cirrhosis. J. Hepatol. 4,723-730 (ref 5); McGovern, B.H., Golan, Y., Lopez, M, et al. (2007) The impact of cirrhosis on CD4+ T cell counts in HIV-seronegative patients. Clin. Infect. Dis. 44(3):431-7 (ref 23); Perrin, D., Bignon, J.D., Beaujard, E., Cheneau, M.L. (1984). Populations of circulating T lymphocytes in patients with alcoholic cirrhosis. Gastroenterol. Clin. Biol. 8(12):907-10]. It had been reported that chronic propranolol treatment can increase the number of circulating T cell [Maisel, A.S., Murray, D., Lotz, M., Rearden, A., Irwin, M., Michel, M.C. (1991). Propranolol treatment affects parameters of human immunity. Immunopharmacology 22(3):157-64]. Pre-surgery administration of propranolol can restore the depletion of CD4+ T helper cell after surgery [Zhou, L., Li, Y., Li, X., et al. (2016) Propranolol Attenuates Surgical Stress-Induced Elevation of the Regulatory T Cell Response in Patients Undergoing Radical Mastectomy. J. Immunol. 196(8):3460-3469]. Above descriptions had been re-emphasized and included in “discussion” of “revised” version [page 15, paragraph 1, line 6-8].

Comment 5: Can the authors present data on the levels of catecholamines (e.g. norepinephrine or as long-standing excessive SNS-activation NPY) ? in other words the expression levels of ADRs and associated changes do relate to the degree of SNS-activity ? although this reviewer is well aware that blood levels do not reflect organ SNS-drive.

Answer 5: It had been reported that increased plasma norepinephrine levels will suppress the activation and function of splenic T cells [Case, A.J., Zimmerman, M.C. (2015) Redox-regulated suppression of splenic T-lymphocyte activation in a model of sympathoexcitation. Hypertension. 65(4):916-923]. Thanks for your comments about the importance of measurement of plasma level of norepinephrine in cirrhotic mice with immune dysfunction.

In comparison with normal animals, higher plasma levels of norepinephrine had been reported in cirrhotic animals [Yangs Y.Y., Lin, H.C., Huang, Y.T., et al. (2002) Effect of 1-week losartan administration on bile duct-ligated cirrhotic rats with portal hypertension. J Hepatol 36(5):600-606;Sansoè, G., Aragno, M., Mastrocola, R., Mengozzi, G., Parola, M. (2016) Alpha-2A Adrenoceptor Agonist Guanfacine Restores Diuretic Efficiency in Experimental Cirrhotic Ascites: Comparison with Clonidine. PloS. One. 11(7):e0158486; Henriksen, J.H., Ring-Larsen, H., Kanstrup, I.L., et al. (1984) Splanchnic and renal elimination and release of catecholamines in cirrhosis. Evidence of enhanced sympathetic nervous activity in patients with decompensated cirrhosis. Gut. 25,1034-1043]. It had been suggested that increased plasma norepinephrine level reflect the sympathetic nerve system (SNS) hyperactivity [Mueller, H.S., Ayres, S.M. (1980) Propranolol decreases sympathetic nervous activity reflected by plasma catecholamines during evolution of myocardial infarction in man. J. Clin. Invest. 65(2):338-346]. In our “revised” version, additional experiments were performed to measure plasma norepinephrine level with stored samples with commercial ELISA kits (Lifespan BioSciences). The results (Table 2) in our study revealed that higher plasma norepinephrine levels (BDL-V: 549±66 pg/mL, TAA-V: 568±95 pg/mL, sham: 298±64 pg/mL) is accompanied by the over-activated splenic SNS and over-expressed splenic ADBRs in cirrhotic mice. Significantly, chronic administration of propranolol in our study decreased the plasma levels of norepinephrine in cirrhotic mice (BDL-V: 549±66 pg/mL; BDL-pro: 398±77 & TAA-V: 568±95 pg/mL; TAA-pro: 398 ±87pg/mL). These results indicated that the beneficial effects of chronic propranolol treatment in cirrhotic mice contributed the suppression of splenic SNS and down-regulated splenic ADBR expression in cirrhotic mice. These results and discussions were included in “revised” version [page 3, line 15; page 5, line 24-25; page 6 (Table 2)].

Comment 6: Results: positive blood cultures in sham animals are not realistic and raise doubts on execution of experiments and in fact, reproducibility.

Answer 6: In table 2, the infection rate was zero in most tissue [blood (anaerobic bacteria), ascites (both aerobic and anaerobic bacteria), lung (aerobic bacteria), intestine (aerobic and anaerobic bacteria)] of sham group indicated the appropriate of our procedure. As mentioned in our “method” section, the sham group are sham-operated mice that had their bile ducts exposed but not ligated. So, the process of bile duct exposed carry risk of infection in some tissue [1/4 in blood (aerobic bacteria), 1/4 of lung (anaerobic bacteria), 1/4 of liver (aerobic and anaerobic bacteria)]. Nonetheless, the infection risk was significantly different between sham and cirrhotic groups.

Comment 7: Interpretation: MNC from peripheral blood have been reported to present with down-regulation of beta-adrenoreceptors in ascitic cirrhotic patients (Gerbes A et al. Lancet 1986). Moreover, Munoz et al. reported an expansion of Th and Teffector cells (Hepatology 2019). The authors should at least discuss this controversy

Answer 7: Thank you very much for your suggestion about two important references, We had read and studied these articles carefully. Bothe our current and previous studies reported the over-expression of ADRB in liver, spleen and on splenic T cells of cirrhotic ascitic mice [ref 11]. For first article (Gerbes A et al. Lancet 1986), which reported that mononuclear cells (MNC) from peripheral blood, although the density or affinity of ADRB2 binding sites are normal, however, the number of binding sites per cell was significantly lower in patients with severe ascites than in patients with mild to moderate or no ascites [Gerbes, A.L., Remien, J., Jüngst, D, et al. (1986) Evidence for down-regulation of beta-2-adrenoceptors in cirrhotic patients with severe ascites. Lancet. 1(8495):1409-1411]. Impaired monocyte function, including defects in chemotaxis, phagocytosis and killing activity, as well as a decrease in the production of lysosomal enzymes, are well-known components of cirrhosis-associated immune dysfunction [Albillos, A., Lario, M., Álvarez-Mon, M. (2014) Cirrhosis-associated immune dysfunction: distinctive features and clinical relevance. J Hepatol. 61(6), 1385-1396; Lancet. 1979 Feb 10; 1(8111):329-30; J Clin Pathol. 1982 Sep; 35(9):972-9]. Our current study focused on evaluation of the expression of ADRB2 on spleen and splenic T cells in ascitic cirrhotic mice. Accordingly, the impacts of chronic propranolol treatment on the expression of ADRB2 of MNC from peripheral blood is needed to be evaluated in future study. Above discussion had been included in “discussion” of “revised” version [page 16, paragraph 4; page 17, paragraph 1].

For second article [Muñoz, L., Borrero, M.‐J., Úbeda, M., et al. (2019) Intestinal immune dysregulation driven by dysbiosis promotes barrier disruption and bacterial translocation in rats with cirrhosis. Hepatology, 70, 925-938]. Munoz et al. reported that the intestinal immune dysregulation driven bacterial translocation in cirrhotic rats. The earlier study from the same group reported that mesenteric Th1 polarization is the first steps to systemic inflammation and bacterial translocation in cirrhotic rats [ref. 7]. The second article that your mentioned reported that expansion of intestinal epithelial and lamina propria lymphocytes T cells and effector T cells subsets are the initiator for bacteria translocation in cirrhosis [Hepatology. 2019, 70, 925-938]. Taken together, it is possible that the expansion of total T/effector T cells induce intestinal inflammation, promote BT, activate and exhausted over-activated splenic T cells and finally immune dysfunction in advanced cirrhosis. Until now, the effector of chronic propranolol treatment on intestinal total T/effector T cells had not yet been explored. So, it is necessary to simultaneously evaluate the effects of chronic propranolol treatment on the intestinal and splenic T cells numbers and subsets in future studies. Above discussion had been included in “discussion” of “revised” version [page 15, paragraph 1, line 14-18].

Comment 8: Presentation: Table 1 does not show what is mentioned in text (page 6/18)

Answer 8: Thanks for your very constructive suggestion about our mixed description of blood, tissue, cell lysates parts in page 6. In “revised” version, we had carefully checked about the description of method. The “original” description about the cell lysates (page 6) part had been moved to page 7.

Comment 9: Fig. 1: symbols for groups not apparent for each subset of images

Answer 9: Thanks for your very constructive suggestion about the missing of the labeling of symbols in each subset of images in Figure 1. In “revised” version, we had added all symbols in each subset of images. Additionally, we had carefully checked the symbols of all figures in “revised” version.

Comment 10: can the authors present a graphical summary on the regulation, modulation and changes associated with cirrhosis in terms of SNS-activity and T-lymphocytes?

Answer 10: Thanks for your very constructive suggestion about including a graphical summary on the regulation, modulation and changes associated with cirrhosis in terms of SNS-activity and T-lymphocytes? We had included that in “revised” version as Figure 8, which is also incorporated into discussion [page 18].

Reviewer 2 Report

The article is interesting, I have some suggestions

1. in tables 2, in Liver weight (mg), is it correct that there is no statistical difference between the groups?
2. In table 2, in Spleen weight (mg), is it correct that there are no statistical differences between the groups?
3. Can you divide figure 1 into two figures?
4. the figure caption of figure 1 is confusing
5. have in triplicate the studies in Figure 1D, 2B and 6A
6. why they use B-actin and GADPH as control. and not just one?
7. The discussion is short. they can expand it

Author Response

Reply to Reviewer 2’s comments

Comment 1: The article is interesting, I have some suggestions; in tables 2, in Liver weight (mg), is it correct that there is no statistical difference between the groups?

Answer 1: Thanks for your reminding about the significance of liver weight between groups. In “revised” version, the significance between sham and BDL; sham and BDL-pro; sham and TAA-V; sham and TAA-pro groups had been included. There are no statistical differences between BDL and BDL-pro; TAA-V and TAA-pro groups, in liver weight.

Comment 2: In table 2, in Spleen weight (mg), is it correct that there are no statistical differences between the groups?

Answer 2: Thanks for your reminding about the significance of spleen weight between groups. In “revised” version, the significance between sham and BDL; sham and TAA-V groups had been included. There are no statistical differences in the spleen weight between sham and BDL-pro; sham and TAA-pro; BDL and BDL-pro; TAA-V and TAA-pro groups.

Comment 3. Can you divide figure 1 into two figures?

Answer 3: Following your suggestion, the “original” figure 1 had been divided as figure 1 and 2 in “revised” version. Further, the numbers of “original” figure 2-6 had been adjusted as figure 3-7 in “revised” version.

Comment 4. the figure caption of figure 1 is confusing

Answer 4: Thanks for your suggestion about the figure caption of figure 1. In “revised” version, the figure caption of 1-2 had been checked and revised. Further, the figure caption of all figures had been checked and modified.

Comment 5. have in triplicate the studies in Figure 1D, 2B and 6A.

Answer 5: Thanks for giving us this opportunity to clarify this point. Figure 1D, 2B and 6A [Fig. 2A,3B,7A] are different experiments. As reported in our methods In order to evaluate the impacts of β-blockers on systemic inflammatory syndrome in cirrhotic mice, the protein expressions of ADRB1 and ADRB2 in various immune dys-regulation-related tissues compared with sham mice. Our initial data revealed that simultaneous increase in ADRB1 and ADRB2 protein levels were noted only in the spleen rather than in the liver, intestine, and MLN (Figure 2A). Thus, the following experiments focused on the spleen (Fig. 3-6). Next, the percentage of CD3+ cells (representing T cells) among all splenocytes or ADRB1+CD3+/ADRB2+CD3+cells among CD3+ splenocytes from different groups of mice were measured by immunofluorescence (IF) staining (Fig 2B). So, original Figure 2B (new Fig 3B), is the measurement of Ki-67 expression among five studied groups. Finally, the original Figure 6A (new Fig 7A) is the protein expression measurements in cell lysates of Th cells isolated from spleen of sham-V or TAA-V mice. The description had been highlighted in text and figure caption [Page 6, paragraph 1, line 4-9]. The labeling in Figures were checked to make them clear.

Comment 6. why they use B-actin and GADPH as control. and not just one?

Answer 6: Both β-actin and GADPH are housekeeping genes that widely used as an internal control to normalize gene and protein expression. In our study, GADPH is used as internal control for various protein expressions in livers, intestine, spleen and mesenteric lymph nodes whereas β-actin is used as internal control for protein expression in cell lysate of splenic Th cells. The reason for used different internal controls was included as below.

          α-Smooth muscle actin (α-SMA) is a hallmark of activated myofibroblasts and has been extensively used to indicate the occurrence and severity of fibrosis in liver diseases. β-actin shares more than 93% sequence identity with α-SMA. There was a positive correlation between the level of β-actin and the content of hydroxyproline in liver. Therefore, the hepatic β-actin level increased significantly during the progression of hepatic fibrosis. This study provides evidences that β-actin is variable and unsatisfied for application as an internal control in hepatic fibrosis [Zhang B, et al. β-Actin: Not a Suitable Internal Control of Hepatic Fibrosis Caused by Schistosoma japonicum. Front Microbiol. 2019 Jan 31;10:66]. Accordingly, it is unsuitable to use β-actin as internal control for evaluation of the protein expression in tissues [liver, spleen, intestine, mesenteric lymph node (MLN)] of cirrhotic mice with hepatic and renal fibrosis with sham controls. On the other hand, it had been reported that GADPH expression stable among rat tissue. Nonetheless, Previous study had shown that beta-actin protein expression are varied among various tissue [Hye Jeong Kim, et al. Evaluation of Protein Expression in Housekeeping Genes across Multiple Tissues in Rats. Korean J Pathol. 2014 Jun; 48(3): 193–200]. Therefore, we used GADPH as internal controls of protein expressions in various tissue.

Comment 7. The discussion is short. they can expand it

Answer 7: Thanks for your detail construction and suggestion. In “revised” version, the length of discussion had extended from 2 pages into 4 phages.

Reviewer 3 Report

Title: Propranolol suppresses the T-helper cell depletion-related immune dysfunction in cirrhotic mice.

In this paper the authors study bacterial translocation (BT) and splenomegaly and cirrhosis in relation to the immune system and beta-blockers. Furthermore, the article focuses attention on the effect of propranolol on the immune, peripheral and splenic systems. The authors conclude that propranolol suppresses ADRB1 and ADRB2 expressions in the spleen and splenic T lymphocytes.

In addition, the treatment improves systemic and splenic immune dysfunction in cirrhosis.

Abstract: The acronyms ADRB1 / ADRB2 must be written in full first.

Fourth last line: Expression should be written with the small letter.

Propranolol acts on the proliferation and apoptosis of endothelial cells in infants and young children. Recently an interesting article has published on this theme. To make this paper more interesting for the readers of this important journal, the authors should expand a little the discussion on this subject, in order to give a wider view to the reader.

Below I list this interesting article that should be studied, the meaning incorporated and reported briefly in the discussion and in the list of references.

Effect of propranolol on proliferation and apoptosis of hemangioma endothelial cells in infants and young children. Xu S, Guo E. J Biol Regul Homeost Agents. 2018 Nov-Dec;32(6):1491-1497.

Again, the immunomodulatory of T helper and Treg in inflammation is very important, but the authors ignore this topic, instead it should be briefly mentioned in the paper, to make it more complete. To make this paper more interesting the authors should expand a little the discussion on this subject. Below I list these interesting articles that should be studied, the meaning incorporated and reported briefly in the discussion and in the list of references.

Immunomodulatory effects of T helper 17 cells and regulatory T cells on cerebral ischemia. Zheng Y, Song T, Zhang L, Wei N. J Biol Regul Homeost Agents. 2018 Jan-Feb;32(1):29-35.

Effects of Astragalus glycoprotein on Th17/Treg cells in mice with collagen-induced arthritis. Wang ZH, Qin C, Ran T, Yang DQ, Guo JH. J Biol Regul Homeost Agents. 2018 Jul-Aug;32(4):951-957.

-I believe these suggestions are important for improving this paper. Without these corrections the paper cannot be published, therefore I recommend minor revision.
I'd like to review this article after corrections.

Author Response

Reply to Reviewer 3’s comments

Comments and Suggestions for Authors: Title: Propranolol suppresses the T-helper cell depletion-related immune dysfunction in cirrhotic mice. In this paper the authors study bacterial translocation (BT) and splenomegaly and cirrhosis in relation to the immune system and beta-blockers. Furthermore, the article focuses attention on the effect of propranolol on the immune, peripheral and splenic systems. The authors conclude that propranolol suppresses ADRB1 and ADRB2 expressions in the spleen and splenic T lymphocytes. In addition, the treatment improves systemic and splenic immune dysfunction in cirrhosis.

 Comment 1: Abstract: The acronyms ADRB1 / ADRB2 must be written in full first.

Answer 1: Thank you very much for your suggestion, We have written ADRB1/ADRB2 in full first in the article as “beta 1 and beta 2 adrenergic receptors” [page 1, abstract, line 8].

Comment 2: Fourth last line: Expression should be written with the small letter.

Answer 2: According to your suggestion, expression was written in small letter [page 1, abstract, line 15]..

Comments 3: Propranolol acts on the proliferation and apoptosis of endothelial cells in infants and young children. Recently an interesting article has published on this theme. To make this paper more interesting for the readers of this important journal, the authors should expand a little the discussion on this subject, in order to give a wider view to the reader. Below I list this interesting article that should be studied, the meaning incorporated and reported briefly in the discussion and in the list of references. Effect of propranolol on proliferation and apoptosis of hemangioma endothelial cells in infants and young children. Xu S, Guo E. J Biol Regul Homeost Agents. 2018 Nov-Dec;32(6):1491-1497.

 Answer 3: Thanks for giving us this important reference and raise the issue that need to be discussed in our text. In vitro studies reported that acute incubation of propranolol promote apoptosis and inhibit the proliferation of cultured endothelial cells (ECs) [Xu, S., Guo, E. (2018) Effect of propranolol on proliferation and apoptosis of hemangioma endothelial cells in infants and young children. J. Biol. Regul. Homeost. Agents. 32(6),1491-1497; Lamy, S., Lachambre, M.P., Lord-Dufour, S., Béliveau, R. (2010) Propranolol suppresses angiogenesis in vitro: inhibition of proliferation, migration, and differentiation of endothelial cells. Vascul. Pharmacol. 53(5-6):200-208]. In cirrhosis, it had been reported that splanchnic, portal, hepatic and pulmonary angiogenesis are mainly contributed to the portal hypertension (PH) and various clinical complications [Serrano, C.A., Ling, S.C., Verdaguer, S., et al. (2019) Portal Angiogenesis in chronic liver disease patients correlates with portal pressure and collateral formation. Dig. Dis. 37(6):498-508]. Systemic angiogenesis induces increased portal inflow and portosystemic collaterals as well as lethal complications of PH such as gastroesophageal variceal hemorrhage (GEV). Propranolol remain the mainstay of pharmacologic treatment for PH and GEV. In cirrhotic portal hypertensive rats, chronic propranolol improve portal hypertension by decreasing in the extent of portal-systemic shunting, splanchnic/hepatic angiogenesis, and liver fibrosis [D'Amico, M., Mejías, M., García-Pras, E., et al. (2012) Effects of the combined administration of propranolol plus sorafenib on portal hypertension in cirrhotic rats. Am. J. Physiol. Gastrointest. Liver. Physiol. 302(10): G1191-G1198]. So, in future study, in addition to the beneficial effects of chronic propranolol treatment on immune dysfunction of cirrhosis in current study, the beneficial effects of chronic propranolol on the parallelly existed cirrhosis-related angiogenesis should be explored. In “revised” discussion, the below discussion had been included [page 17, paragraph 4].

Comments 4: Again, the immunomodulatory of T helper and Treg in inflammation is very important, but the authors ignore this topic, instead it should be briefly mentioned in the paper, to make it more complete. To make this paper more interesting the authors should expand a little the discussion on this subject. Below I list these interesting articles that should be studied, the meaning incorporated and reported briefly in the discussion and in the list of references. Immunomodulatory effects of T helper 17 cells and regulatory T cells on cerebral ischemia. Zheng Y, Song T, Zhang L, Wei N. J Biol Regul Homeost Agents. 2018 Jan-Feb;32(1):29-35. Effects of Astragalus glycoprotein on Th17/Treg cells in mice with collagen-induced arthritis. Wang ZH, Qin C, Ran T, Yang DQ, Guo JH. J Biol Regul Homeost Agents. 2018 Jul-Aug;32(4):951-957.

Answer 4: Thanks for giving us these important references and raising the issue that need to be discussed in our text. Under normal circumstances, Th17 and Treg cells maintain a dynamic balance and result in the induction of immune responses of appropriate intensity, which is conducive to the maintenance of a stable immune state in the body. It had been reported that the degree of disequilibrium between Th17 and Treg level is correlated with the severity of systemic inflammation in mice with post-cerebral ischemia and collagen-induced arthritis [Biol Regul Homeost Agents. 2018 Jan-Feb;32(1):29-35; J Biol Regul Homeost Agents. 2018 Jul-Aug;32(4):951-957]. Treg/Th17 imbalance is closely related to many immune disorders and infectious diseases [Kleinewietfeld, M., Hafler, D.A. (2013) The plasticity of human Treg and Th17 cells and its role in autoimmunity. Semin. Immunol. 25(4):305-312]. Treg/Th17 imbalance is involved in the pathogenesis of liver cirrhosis and predicting the decompensation of liver cirrhosis [Lan, Y.T. (2019) Treg/Th17 imbalance and its clinical significance in patients with hepatitis B-associated liver cirrhosis. Diagn. Pathol. 2019; 14: 114]. T lymphopenia and immune dysfunction usually occur in patients with advanced cirrhosis, whose are characterized by abnormal hepatic neo-angiogenesis, hepatic ischemia and systemic inflammation [J. Hepatol. 2013;4,723-730; Hepatology.2005;2,411-419; Dig. Dis. 2019, 37(6):498-508; Am. J. Physiol. Gastrointest. Liver. Physiol. 302(10): G1191-G1198]. In current study, we mainly explore the effects of chronic propranolol treatment on the abnormal changes in the Th1, Th2, Treg cytokines in mice with cirrhosis-related immune dysfunction. So, in future study, it is necessary to explore the effects of chronic propranolol treatment on imbalance between Th17 and Treg cell on the cirrhosis-associated immune dysfunction. In “revised” discussion, the above discussion had been included [page 17, paragraph 1].

Comments 5: I believe these suggestions are important for improving this paper. Without these corrections the paper cannot be published, therefore I recommend minor revision.

Answer 5: We are very appreciating for your very constructive comments about our manuscript. I have detailly reviewed your suggested important articles and discussed the context in our study. The revision really makes our article complete and more worth reading.

Round 2

Reviewer 2 Report

The authors did a good review work. Congratulations